# Analysis of travel-time to HIV treatment in sub-Saharan Africa reveals inequities in access to antiretrovirals
Justin T. Okano [1], Andrea Low[2], Luckson Dullie[3], Wongani Mzumara[4],
Harriet Nuwagaba-Biribonwoha[2] & Sally Blower [1] ✉

## Abstract

**Background** UNAIDS proposes ending inequalities in access to HIV treatment. We use data from nationally-representative Population-Based HIV Impact Assessment (PHIA) surveys for Eswatini, Malawi, and Zambia to identify inequities in one-way travel-time to access antiretroviral therapy (ART) for people with HIV (PWH).
**Methods** Using biometric data from the PHIAs, we construct Epidemic Surface Prevalence maps and estimate treatment coverage. Self-reported travel-time data were fit using logistic cumulative distribution functions. Multivariable logistic regression models were used to examine relationships between travel-time, urban-rural residency, age, and sex.
**Results** We find the majority of PWH on ART are women: Eswatini (69.4%), Malawi (64.8%), Zambia (63.0%). The majority on ART reside in rural areas in Malawi (74.6%) and Eswatini (71.0%), but in urban areas in Zambia (61.9%). Travel-time distribution functions show, on average, PWH in Eswatini have the shortest travel-times; travel-times in Malawi are slightly longer than in Zambia. 56.4% (Malawi), 50.5% (Zambia), and 37.4% (Eswatini) of treated individuals could not access ART within one hour; many travel more than two hours: 20.6% (Zambia), 19.0% (Malawi), 10.5% (Eswatini). In all countries, the odds of traveling one or more hours are significantly higher in rural than urban areas. In Eswatini and Zambia, women have significantly higher odds than men of traveling one or more hours.
**Conclusions** Many PWH spend considerable time traveling to access ART. Substantial inequities exist, disadvantaging rural populations in all three countries, and women in Eswatini and Zambia. Achieving UNAIDS' goal will require identifying drivers of inequities and designing strategies to minimize them.

## Plain language summary

UNAIDS recently announced a new goal: to end inequalities in accessing HIV treatment. We used data from the nationally-representative Population-Based HIV Impact Assessment surveys for Eswatini, Malawi, and Zambia to identify inequities in one-way travel-time to treatment. We found considerable inequities. Patients in Eswatini (on average) had the shortest travel-times; patients in Zambia had slightly shorter travel-times than patients in Malawi. 56.4% (Malawi), 50.5% (Zambia), and 37.4% (Eswatini) of patients could not access ART within one hour; many traveled for more than two hours: 20.6% (Zambia), 19.0% (Malawi), and 10.5% (Eswatini). On average, it took longer to access ART in rural (versus urban) areas. Women in Eswatini and Zambia (on average) spent more time traveling to access treatment than men. To achieve UNAIDS' goal, strategies that minimize inequities in travel-time need to be designed.

UNAIDS has recently announced a new strategy: End Inequalities. End AIDS. Global AIDS Strategy 2021-2026[1]. The goal is to eliminate HIV by achieving a high treatment coverage in an equitable manner. In order for people with HIV (PWH) to utilize HIV treatment, it is necessary for them to travel to access treatment at healthcare facilities (HCFs). However, it is currently unknown – at the national-level – how long PWH have to travel to access antiretroviral treatment (ART), and if inequities exist in travel-time. Travel-time to treatment is important. Studies in sub-Saharan Africa (SSA) have found that long distances/travel-times have a negative impact on

engagement in HIV services, HIV screening and testing regularity, adherence to ART and pre-exposure prophylaxis (PrEP) regimens, and can also increase mental health issues[2–13]. A study of those lost to follow-up from HIV treatment programs in six countries in SSA has found that death rates were the highest among those who had long (self-reported) travel-times to reach HCFs[14].

The impact of travel-time to access ART on HIV care varies between urban and rural settings. Notably, rural areas are underserved in all aspects of care, including access to specialized services, and advanced laboratory

[1]Center for Biomedical Modeling, Semel Institute for Neuroscience and Human Behavior, David Geffen School of Medicine, University of California Los Angeles, Los Angeles, USA. [2]ICAP at Columbia University, Mailman School of Public Health, Columbia University, New York, USA. [3]Partners In Health/Abwenzi Pa Za Umoyo, Neno, Malawi. [4]Department of HIV and AIDS, Ministry of Health Malawi, Lilongwe, Malawi. Grants: National Institute of Allergy and Infectious Diseases (R01 AI167713: J.T.O., L.D., W.M., and S.B.). ✉e-mail: sblower@mednet.ucla.edu

testing and imaging[15]; furthermore, individuals in rural areas often lack privacy with respect to medical issues as they live in small communities. In SSA, health professional shortages are especially pronounced in rural areas[15,16]. Rural communities are also more prone to poverty, making long travel-times to HCFs particularly unsustainable[17,18]. Rural residence has been found to be associated with less HIV testing[19], less adherence to ART[20], and lower rates of viral load suppression; the latter is in part due to lack of access to regular viral load testing[21]. A systematic review of 30 studies in SSA has also found that rural PWH had a two-fold higher odds of being lost to follow-up from HIV treatment programs[22].

The urban-rural health gap can be complicated by the sex imbalance in population distribution, with men tending to be more numerous in cities due to job opportunities and family constraints on women[23]. Men are likely to have greater economic ability to access healthcare. However, men (in comparison with women) continue to have higher mortality from HIV, including when they are on ART[24,25]. Many studies have also shown that men are more likely to present with late diagnosis of HIV[25,26], with much of the delay in men accessing HIV treatment being attributable to lack of diagnosis, and less frequent uptake of testing[27]. Even in urban areas, men living with HIV are less likely (than women) to be aware of their HIV-status[28].

Here, we analyze questionnaire and biometric data from the nationally-representative Population-Based HIV Impact Assessment (PHIA) surveys[29] conducted in Eswatini, Malawi, and Zambia in order to estimate the travel-time needed to access ART, and to identify any potential inequities in travel-times. Since 2015, PHIA surveys have been conducted in many of the countries that are most affected by the HIV pandemic; these nationally representative cross-sectional surveys assess the current status and effectiveness of national programs in achieving epidemic control. They collect data on HIV-testing, uptake of HIV services, and adherence to ART[29]. We use these data to address the following questions: How long does it take PWH in Eswatini, Malawi, and Zambia to travel to access ART? Do inequities exist in travel-time to ART? If so, are these country-specific? Are they associated with the area of residency (urban or rural) and/or sex?

Our results show that many PWH in Eswatini, Malawi, and Zambia spend several hours traveling for ART, however, there are considerable inequalities in travel-time. We find that it takes substantially longer to access ART in rural areas than in urban areas, and that women in Eswatini and Zambia spend more time traveling to access ART than men.

## Methods
### Datasets
The PHIA data were collected in 2016/17 in Eswatini, 2015/16 in Malawi, and 2016 in Zambia[29]. The surveys used a two-stage stratified cluster sampling design. In the first stage, survey clusters (i.e., enumeration areas; EAs) were selected with probability proportional to strata delineated by the most recent census: 286 EAs were selected in Eswatini, 500 EAs in Malawi, and 511 EAs in Zambia. In the second stage, households were randomly sampled from each EA. All individuals are nested within georeferenced EAs, which are geomasked to ensure de-identification[30]; we analyzed anonymized data.

PHIA participants were 15 years and older who were de facto residents of a household (residents and visitors who had slept at the household the night before the survey). All participants provided informed consent. This was obtained by describing to participants (in a language they understood) the study's objectives, protocol, risks and benefits, and their rights, specifically the right to withdraw at any time without penalty. They then provided an electronic signature on a data collection tablet; for participants with limited literacy, a literate witness was present to ensure their comprehension.

Each PHIA survey was designed to measure key biological outcomes using a nationally-representative sample; samples reflected the socio-demographic characteristics of the population (e.g., sex, age, urban-rural distribution). To do so, the surveys collected questionnaire data on socio-demographic variables and biometric data (i.e., blood samples). The biometric data were used to identify the HIV status of each participant and to detect the presence of antiretroviral medications. HIV status was identified by using a serological rapid diagnostic testing algorithm. An individual was defined as a PWH if antibodies to HIV were detected in their blood; this definition did not depend upon viral load. Following WHO guidelines, all HIV-positive samples underwent confirmatory testing using the Geenius HIV 1/2 Supplementary Assay, and extensive quality assurance testing. Each participant was asked on the questionnaire to self-identify as to whether they were male or female, and to report their HIV status. Participants who reported that they were HIV-positive (i.e., those who were aware of their diagnosis) were asked if they were on treatment. Participants who answered in the affirmative were then asked to specify, based on their most recent HIV care visit, their one-way travel-time to reach the HCF they used for ART. They were asked to choose from one of three categories: less than one hour, one to two hours, or more than two hours. Therefore, all travel-time data were self-reported and collected via the questionnaire. Individuals' questionnaire responses were linked to their biometric data.

Among adults selected to participate in PHIA, the individual survey response rate was 91% in Eswatini, 87% in Malawi, and 86% in Zambia. Response rates were higher in women than men: 94% vs. 87% in Eswatini, 93% vs 82% in Malawi, and 91% vs. 80% in Zambia. Biomarker testing was offered to all individuals who completed the individual survey. Women were only marginally more likely to provide blood samples than men: 95% vs 92% in Eswatini, 88% vs 87% in Malawi, and 91% vs 89% in Zambia.

### Ethical considerations
The PHIA study in Eswatini was cleared by Institutional Review Boards (IRBs) at the Center for Disease Control and Prevention (#6846), Columbia University Medical Center (IRB-AAAQ8889), Westat (#6420), and by the Swaziland Scientific and Ethics Committee (MH/599C/FWA 000 15267/IRB 000 9688). The PHIA study in Malawi was cleared by IRBs at the Center for Disease Control and Prevention (#6692), Columbia University Medical Center (IRB-AAAO9051), Westat (#6264), and by the National Health Sciences Research Committee in Malawi (NHSRC #1361). The PHIA study in Zambia was cleared by IRBs at the Center for Disease Control and Prevention (#6760), Columbia University Medical Center (IRB-AAAQ0753), Westat (#6317), and by the Tropical Diseases Research Center Ethics Committee (TRC/C4/18/2015) and National Health Research Authority (MH/101/23/10-1). For all studies, guidelines for Strengthening the Reporting of Observational Studies in Epidemiology (STROBE) were followed.

### Inclusion criteria
The inclusion criteria for our study were all de facto residents 15 to 59 years old who provided a blood test; this resulted in a sample of $N = 9559$ individuals in Eswatini, $N = 16,700$ individuals in Malawi, and $N = 19,115$ individuals in Zambia. Participant demographics for the three countries in our study are presented in Table 1. The country-specific biometric data from the PHIA surveys were used to estimate HIV prevalence. Therefore, prevalence estimates are based on all individuals in the PHIA study who had detectable HIV antibodies in their blood: i.e., PWH who were aware that they were infected with HIV before the PHIA survey, and PWH who were unaware that they were infected with HIV before the PHIA survey. The country-specific biometric data from the PHIA surveys were also used to estimate treatment coverage; coverage is defined as the percentage of PWH (aged 15–59) in the PHIA study who, based on the detection of anti-retrovirals, were on treatment. Treatment coverage estimates were based on 2785 PWH in Eswatini, 2147 PWH in Malawi, and 2439 PWH in Zambia. 93% of individuals on ART answered the question on travel-time to the nearest clinic; for these individuals, we have complete data on sex, urban-rural status, and age.

### Statistical analysis
For national-level estimation of HIV prevalence and treatment coverage, we made use of PHIA survey weights, which account for selection probabilities and non-response. Sampling errors were calculated in Stata (v.16)[31] using

**Table 1 | PHIA participant demographics for Eswatini, Malawi, and Zambia**

| | Eswatini | | Malawi | | Zambia | |
|---|---|---|---|---|---|---|
| | (N = 9559) | | (N = 16,700) | | (N = 19,115) | |
| **Age** | | | | | | |
| 15–24 | 3601 | (37.2%) | 6258 | (39.8%) | 7320 | (41.1%) |
| 25–34 | 2601 | (29.0%) | 4744 | (27.6%) | 5133 | (27.3%) |
| 35–44 | 1700 | (18.7%) | 3256 | (18.2%) | 3736 | (18.1%) |
| 44–59 | 1657 | (15.1%) | 2442 | (14.4%) | 2926 | (13.5%) |
| **Sex** | | | | | | |
| Men | 4023 | (45.5%) | 6956 | (48.5%) | 8142 | (49.0%) |
| Women | 5536 | (54.5%) | 9744 | (51.5%) | 10,973 | (51.0%) |
| **Residence** | | | | | | |
| Rural | 7428 | (72.0%) | 10,390 | (79.9%) | 10,775 | (54.3%) |
| Urban | 2131 | (28.0%) | 6310 | (20.1%) | 8340 | (45.7%) |

The age distribution, sex distribution, and urban-rural residency of study participants from the 2016/17 PHIA survey in Eswatini, the 2015/16 PHIA survey in Malawi, and the 2016 PHIA survey in Zambia. Individuals had to be 15 to 59 years old and have provided a blood test. The table shows the sample size and weighted estimates of the percentage in each group.

jackknife replicate weights which account for PHIA's complex sampling design[32]. For between-group comparisons of prevalence or treatment coverage, we used two-tailed F-Tests evaluated at α = 0.05.

The self-reported travel-time data were fit using logistic cumulative distribution functions (CDFs). Data were then stratified based on urban-rural residency, and 95% confidence bands were calculated.

Country-specific multivariable logistic regression models were fit to examine the relationship between traveling for one or more hours to access ART and urban-rural status, sex, and age. We calculated age-adjusted Odds Ratios (aORs) and 95% confidence intervals (CIs) to assess the effect of each predictor; all models were fit in R (v. 4.1.2)[33]. All results on travel-time refer to the one-way travel-time to the HCF, not to round-trip travel-time.

### Epidemic Surface Prevalence maps
We created HIV Epidemic Surface Prevalence (ESP) maps for Eswatini, Malawi, and Zambia. We did not create treatment coverage maps, as the small sample size of infected individuals at each cluster site would have led to a high degree of uncertainty.

To construct the ESP maps, we used Empirical Bayesian kriging[34] (EBK) to spatially interpolate the country-specific cluster-level estimates of HIV prevalence from the PHIA data. Kriging is a spatial interpolation technique that utilizes the spatial correlation structure of data to predict the value of a variable at any point in space[35]. Kriging has been widely used in spatial epidemiology[36–40], and shown to be fairly robust to the geographic displacement of sampling sites[41], which occurred during the geomasking of the PHIA data. We evaluated the validity of the EBK model for each country, by using cross-validation (CV)[42,43] to compute several well-established metrics. During CV, each point is sequentially removed, and the remaining points are used to predict HIV prevalence. The estimated values can then be compared with the predicted values to adjudge model accuracy and assess the quality of each model's predictions. The ESP maps show the mean values of the interpolated estimates. We used R (v. 4.1.2)[33] for data processing and ArcGIS Pro (v. 3.2.2)[44] for EBK.

### Density of infection maps
Country-specific Density of Infection (DoI) maps were created by using raster (i.e., grid) multiplication to combine each of the HIV ESP maps with contemporaneous gridded population data (for 15 to 59-year-olds) from the WorldPop database[45]; we have previously created HIV DoI maps for several countries in SSA[36,46,47]. WorldPop data are demographic data of population density that are updated annually to reflect UNAIDS' projected urban-rural growth rates; we aggregated the WorldPop data to a resolution of one square

kilometer. DoI maps reveal the number of PWH per km$^2$. We used R (v. 4.1.2)[33] for raster multiplication and ArcGIS Pro (v. 3.2.2)[44] for mapping.

### Epidemic concentration curves
We used the country-specific DoI maps to estimate the number of PWH in each square kilometer in each country. Each set of country-specific estimates was then ranked and plotted on a semi-logarithmic scale; each plot is an Epidemic Concentration Curve (ECC)[46]. The ECC shows the minimum DoI level that would need to be on treatment to achieve different treatment coverages.

### Reporting summary
Further information on research design is available in the Nature Portfolio Reporting Summary linked to this article.

## Results
### HIV prevalence
Eswatini has the most severe HIV epidemic (27.9% prevalence [95% CI: 26.6–29.3%]); HIV prevalence was similar in Malawi (10.5% [95% CI: 10.0–11.2%]) and Zambia (12.0% [95% CI: 11.4–12.7%]). Not surprisingly, in all three countries, HIV prevalence was significantly higher in women than men (Table 2). Hence, the majority of PWH were women: 66.8% in Eswatini, 61.0% in Malawi, and 62.2% in Zambia. In Malawi and Zambia, HIV prevalence was significantly higher in urban areas than in rural areas (Table 2). We found that Zambia has a predominantly urban epidemic (only 41.8% of PWH are residents of rural areas), whereas Malawi and Eswatini have predominantly rural epidemics (73.1% and 69.6%, respectively, of PWH are residents of rural areas).

For all three countries, the HIV ESP maps (Fig. 1) show that there are large-scale geospatial patterns in HIV prevalence that cross administrative boundaries (compare Fig. 1 with Supplementary Fig. 1), and that there is considerable geographic variation in prevalence: 14.4–53.2% in Eswatini, 1.4–26.2% in Malawi, and 1.0–31.6% in Zambia. Sex-specific ESP maps (Fig. 2) show that the geospatial patterns in HIV prevalence differ between women and men. In agreement with previous studies in SSA[48,49], HIV prevalence is generally higher in women than in men. There are some areas in Malawi and Zambia where HIV prevalence in men is predicted to be higher than in women (Supplementary Fig. 2); there are no such areas in Eswatini. The cross-validation metrics indicate that the EBK models are unbiased and properly configured (Supplementary Table 1). The maximum DoI for each country (Fig. 3a–c) occurs in a city (compare Fig. 3a-c with Supplementary Fig. 1): 3414 PWH/km$^2$ in Malawi, 1270 PWH/km$^2$ in Zambia, and 425 PWH/km$^2$ in Eswatini. This high DoI reflects both the high population density in urban areas, and the fact that (in general) HIV prevalence is greater in urban areas in comparison with rural areas. Notably, in all three countries, there are some areas where there are unlikely to be any PWH.

### Treatment coverage
Treatment coverage was significantly higher (P < 0.001) in Eswatini (76.6% [95% CI: 74.8–78.4%]) than in Malawi (70.0% [95% CI: 67.5–72.3%]); coverage in Malawi was significantly higher (P < 0.001) than in Zambia (62.6% [95% CI: 60.0–65.1%]). In Eswatini and Malawi, coverage was significantly higher in women than in men; there was no significant difference in Zambia (Table 2). The majority of PWH on ART were women: 69.4% in Eswatini, 64.8% in Malawi, and 63.0% in Zambia. Both Eswatini and Malawi had significantly higher treatment coverage in rural areas than in urban areas (Table 2), and the majority of PWH on ART reside in rural areas (71.0% and 74.6%, respectively). In contrast, Zambia had higher treatment coverage in urban areas than in rural areas (Table 2), and the majority (61.9%) of PWH on ART reside in urban areas. Notably, in all three countries, ART programs were treating some PWH at a very low DoI (Fig. 3d–f).

### Travel-time inequities
Substantial inequities in travel-time to access ART were found amongst, and within, the three countries (Fig. 4a). On average, PWH in Eswatini reported

**Table 2 | Sex-specific and urban-rural differences in HIV prevalence and treatment coverage**

| HIV prevalence | Men | | Women | | F | P | Rural | | Urban | | F | P |
|---|---|---|---|---|---|---|---|---|---|---|---|---|
| Eswatini | 20.4% | (4023) | 34.2% | (5526) | 307 | <0.001* | 27.0% | (7428) | 30.4% | (2131) | 2.41 | 0.133 |
| Malawi | 8.5% | (6956) | 12.5% | (9744) | 67.0 | <0.001* | 9.6% | (10,390) | 14.1% | (6310) | 53.3 | <0.001* |
| Zambia | 9.3% | (8142) | 14.6% | (10,973) | 162 | <0.001* | 9.3% | (10,775) | 15.3% | (8340) | 78.5 | <0.001* |
| **Treatment coverage** | **Men** | | **Women** | | **F** | **P** | **Rural** | | **Urban** | | **F** | **P** |
| Eswatini | 70.7% | (869) | 79.6% | (1916) | 30.3 | <0.001* | 78.3% | (2079) | 72.9% | (706) | 7.67 | 0.010* |
| Malawi | 63.1% | (677) | 74.3% | (1470) | 24.1 | <0.001* | 71.4% | (1119) | 66.1% | (1028) | 5.64 | 0.026* |
| Zambia | 61.6% | (767) | 63.2% | (1672) | 0.54 | 0.468 | 57.6% | (1037) | 66.2% | (1402) | 11.8 | 0.002* |

HIV prevalence and treatment coverage estimates are given as weighted percentages, with the sample size (n) they are calculated from. Hypothesis tests for differences in A) HIV prevalence and B) treatment coverage between (i) men and women and (ii) rural and urban residents were two-tailed and evaluated at α = 0.05; significant differences are denoted with an asterisk (*).

the shortest travel-times (green data); reported travel-times in Malawi (pink data) were slightly longer than in Zambia (purple data). A high proportion of PWH in all three countries reported traveling for one or more hours to access ART (Table 3). Notably, a fairly high percentage reported needing to travel for more than two hours to access ART.

In all three countries, we found substantial urban-rural inequities in travel-time to access ART (Fig. 4b–d). The functions for the urban areas (red curves) are steeper than for the rural areas (blue curves). This signifies that, on average, PWH in urban areas had the shortest travel-times to access ART. The 95% confidence bands provide a measure of model uncertainty: the standard errors of the fitted models range from small-to-moderate (for Eswatini and Malawi) to moderate-to-large (for Zambia). In rural areas, a higher proportion of PWH reported traveling for one or more hours to access ART than in urban areas (Table 3). We found a similar effect for PWH who reported traveling for over two hours to access ART. However, even in urban areas, quite a few PWH traveled for over two hours to access ART.

In Eswatini and Malawi, rural residents have greater than two times the odds of traveling for one or more hours to access ART than urban residents (Table 4). In Zambia, rural residents have greater than three times the odds of traveling for one or more hours for ART than urban residents. Additionally, in Eswatini and Zambia, women had significantly higher odds of traveling for one or more hours to access ART than men (Table 4). However, this was not the case in Malawi, where women were not more likely than men to travel for one or more hours for ART. We did not find an interaction effect between urban-rural residence and sex – on the odds of traveling for one or more hours to access ART.

## Discussion

Eswatini, Malawi, and Zambia have many similarities in their HIV epidemics. They all show considerable geographic variation in both HIV prevalence and DoI, prevalence is higher in women than in men, and greater in urban areas than in rural areas, and the majority of PWH (and the majority of PWH on ART) in each country are women. However, Eswatini and Malawi have a predominantly rural epidemic, while Zambia has a predominantly urban epidemic: in Eswatini and Malawi the majority of PWH (and the majority of PWH on ART) reside in rural areas, whereas, in Zambia the majority of PWH (and the majority of PWH on ART) reside in urban areas. We found that many PWH in all three countries spent a considerable amount of time traveling to access ART; this may be due to the fact that, in all three countries, the DoI is fairly low in many areas. At a low DoI, the population density of PWH is low and dispersed. There are relatively few HCFs providing ART in areas with these demographic conditions. Hence, a low DoI generally results in long travel-times to access ART. Our study shows that there are considerable inequalities amongst and within countries in the travel-time needed to access ART. We have found, not surprisingly (due to the scarcity of HCFs in rural areas), that it takes longer on average to access ART in rural areas than in urban areas, but surprisingly – in all three countries – some PWH living in urban areas need to travel for several hours to access ART. In addition, we have found that women in Eswatini and Zambia, on average, spend more time traveling to access ART than men.

The travel-time distributions that we have found reflect, to some degree, the geographic distribution of the population relative to the geographic distribution of HCFs: i.e., population density is likely to be greatest around HCFs, and decrease with distance. However, the travel-time distributions are also likely to reflect, to some degree, the distance decay effect; i.e., the further away an individual lives from a HCF, the less likely they are to access healthcare[50]. If a distance decay effect is occurring, some PWH may not be accessing ART because of the long time that they would have to spend traveling. Throughout Africa, long travel-times have been found to be a barrier for antenatal care[51], fever-seeking behavior[52], and HIV care[2,53].

The country-specific differences in travel-time that we have found may be due to epidemiological factors (e.g., epidemic severity, and geographic variation in prevalence), geographic factors (e.g., topography and land cover), demographic factors (e.g., population distribution and density), the healthcare system (e.g., geographic distribution of healthcare facilities, their

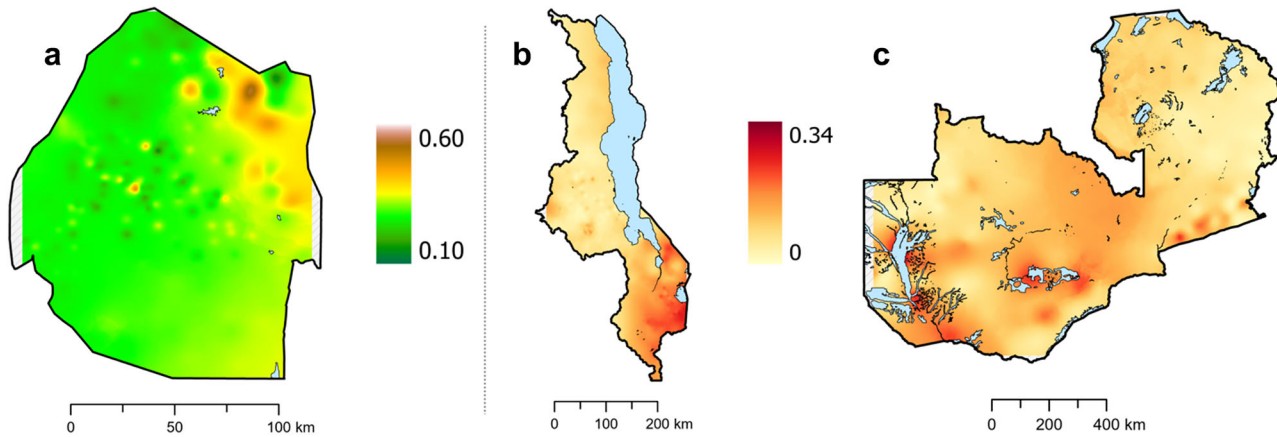

**Fig. 1 | HIV Epidemic Surface Prevalence (ESP) maps.** Shown for **a** Eswatini, **b** Malawi, and **c** Zambia. The first color scale refers to Eswatini, the second color scale refers to both Malawi and Zambia. The maps are not plotted on the same scale because the range for HIV prevalence for Eswatini is much larger than the range for HIV prevalence for Malawi and Zambia.

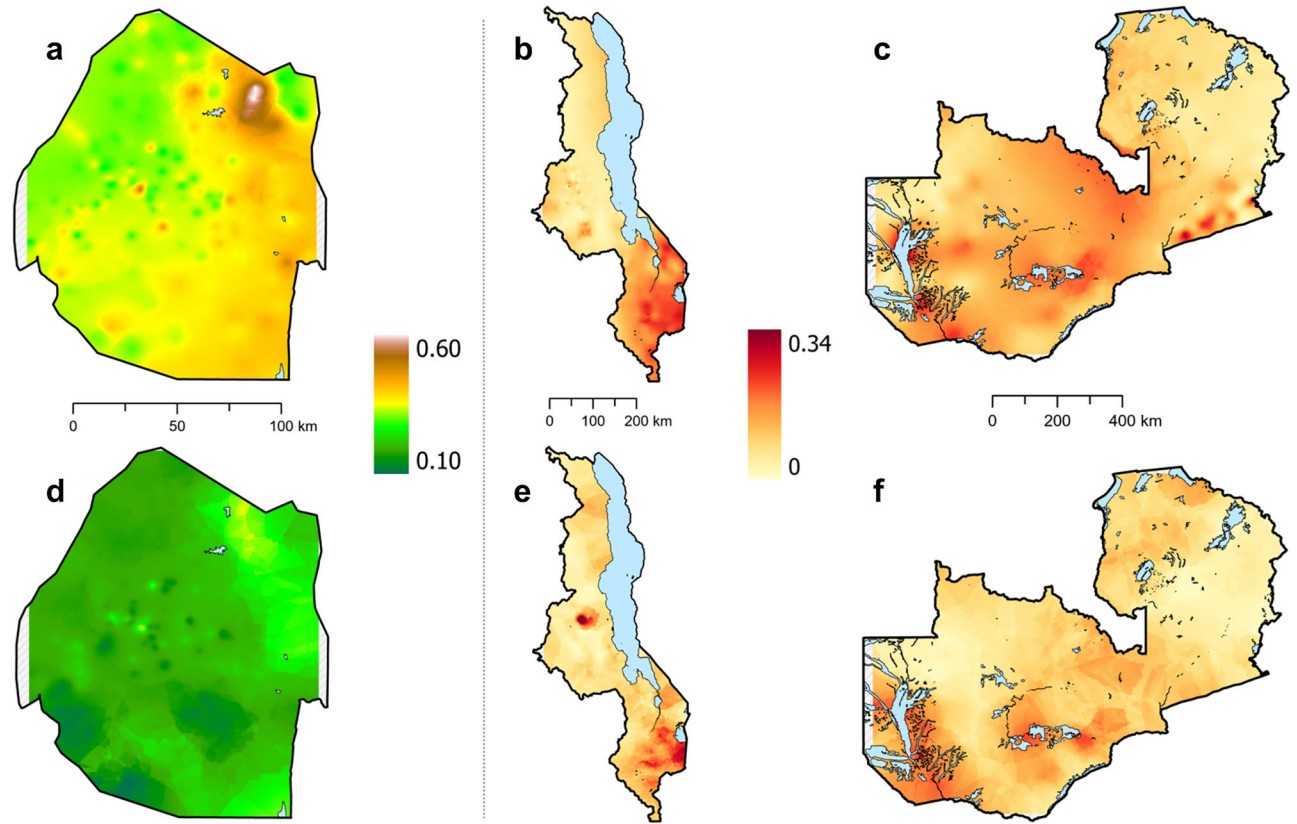

**Fig. 2 | Sex-specific ESP maps.** Shown for **a** Women in Eswatini, **b** Women in Malawi, **c** Women in Zambia, **d** Men in Eswatini, **e** Men in Malawi, and **f** Men in Zambia. The first color scale refers to Eswatini, and the second color scale refers to both Malawi and Zambia. The maps are not plotted on the same scale because the range for HIV prevalence for Eswatini is much larger than the range for HIV prevalence for Malawi and Zambia.

number, and quality), transportation (e.g., availability, and structure of the transportation network), socio-behavioral factors (e.g., stigma), and treatment coverage (e.g., overall level of coverage, and geospatial variation in coverage). Notably, Eswatini has both the highest coverage of treatment and the shortest average travel-time to access ART.

The considerable within-country variation in travel-time that we have found may be due to the same factors that result in country-specific differences. However, we have identified two factors that enable us to understand, to some degree, the within-country heterogeneity in travel-time: urban-rural differences, and (in the case of Eswatini and Zambia) sex differences. In all three countries, the average travel-time to access ART is highest in rural areas and lowest in urban areas. This result is understandable as – in all three countries – the DoI is highest, and HCFs are more concentrated, in urban areas. We have also found that women and men have different utilization patterns with respect to ART: in all three countries women are more likely to use ART than men, despite the fact that in Eswatini and Zambia the travel-time needed to access ART is (on average) longer for women than for men. The reason for the differences between the sexes is not obvious. There are at least three possible explanations. One is that there may be more women, than men, with HIV and on ART in rural areas. A second possibility is that women may be more likely to demonstrate bypass behavior than men: i.e., some women may choose, for various

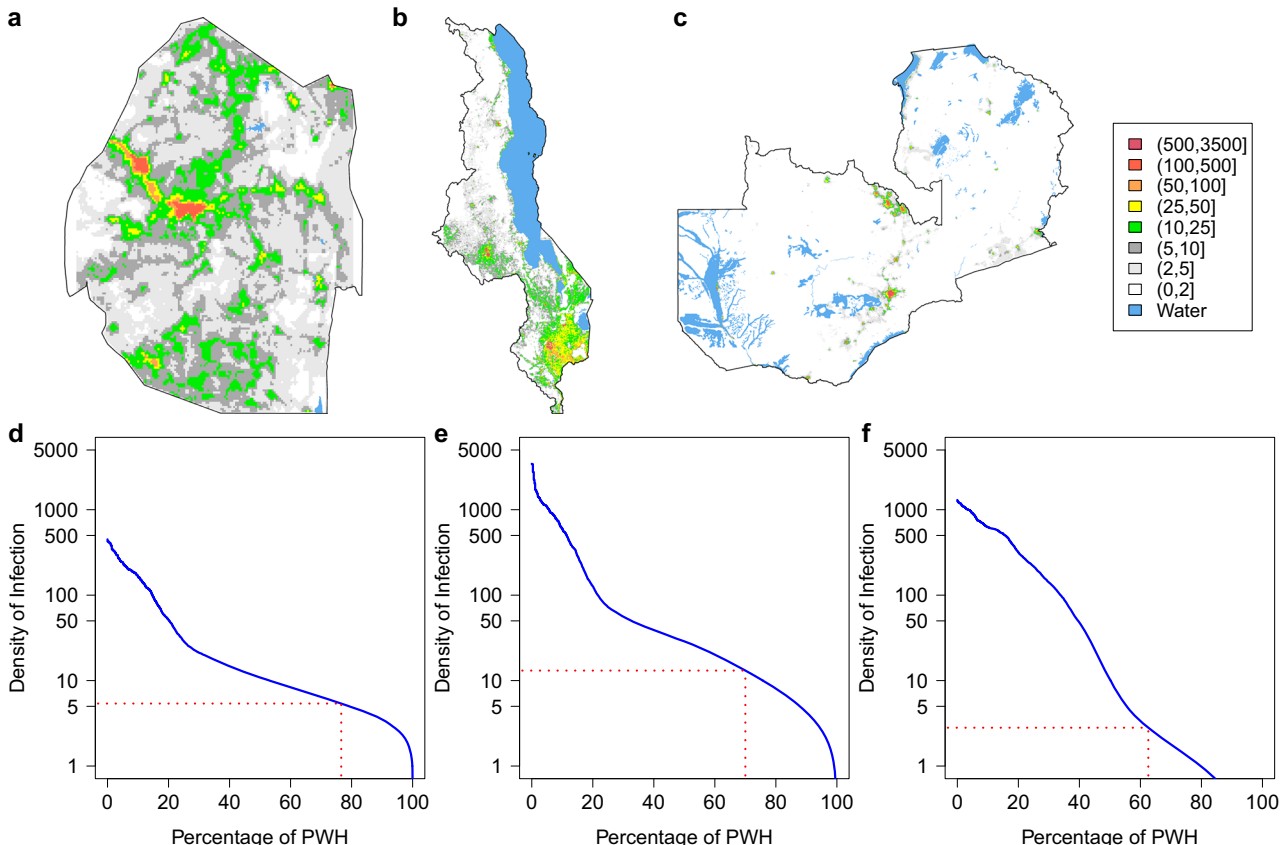

**Fig. 3 | Density of Infection (DoI) maps and Epidemic Concentration Curves (ECCs).** DoI maps for **a** Eswatini, **b** Malawi, and **c** Zambia showing the number of PWH/km². ECCs for **d** Eswatini, **e** Malawi, and **f** Zambia. The red dotted lines denote the minimum DoI that would need to be on treatment to achieve each country's current treatment coverage (estimated at the time their PHIA data were collected); this corresponds to 13.1 PWH/km² in Malawi, 5.4 PWH/km² in Eswatini, and 2.8 PWH/km² in Zambia.

reasons, not to utilize their nearest HCF[54]. Finally, distance decay functions may be sex-specific.

In our study, we chose to analyze self-reported data on how long it took PWH (who participated in the PHIA study) to reach the HCF they used to receive their medications; i.e., we used reported travel-times. We[47] and others[55–58] have previously estimated travel-times to HCFs by using a combination of modelling, friction surface maps, geo-located HCFs, and gridded population data. These modelled travel-times can vary as a function of the database that is used for the gridded population data[56]. In order to model travel-times, two assumptions are made: individuals take the most direct route, and they use their nearest HCF. However, these assumptions do not always hold. Notably, it has been shown (using aggregated and anonymized smartphone data from 100 countries) that revealed travel-times (which may be expected to be similar to reported travel-times) to HCFs were generally longer than modelled travel-times[57]; this may be because individuals did not take the most direct route and/or did not use their nearest HCF. Another study, which focused on the geographic accessibility of family planning outlets in Kenya, compared women's self-reported travel-times with modelled travel-times[58]. It was found that reported travel-times were longer than modelled travel-times. Additionally, only 52% of women used their nearest family planning outlet, even if their chosen method for contraception was available; this fell to ~20% if their chosen method was unavailable.

Inequalities in access to healthcare, including inequalities in travel-time to reach HCFs are not limited to HIV-related care[55,56]. Global maps of the travel-time needed for the general population to reach the nearest HCF to access healthcare have recently been constructed by modelling travel-time[55]. These maps have highlighted and quantified the difficulty poorer individuals in remote areas have in accessing HCFs, and the importance of transportation in increasing access. Weiss and colleagues[55] have shown that 9% of the global population cannot reach their nearest HCF within one hour (even if they have access to motorized transportation), and 43% cannot reach their nearest HCF within one hour if walking. The PHIA surveys did not collect any data on the type of transportation that PWH used to travel to HCFs, but the majority of individuals in SSA walk to access healthcare. Our results show that over one-half of PWH in Malawi and Zambia and over one-third of PWH in Eswatini could not reach an HCF within one hour (Table 3). Our results are based on reported travel-times, and therefore cannot be directly compared with the previous estimates[55] which are based on modelled travel-times; reported and modelled travel-times are based upon different assumptions[57,58].

Population growth and increasing urbanization in all African countries are likely to have an effect on the travel-time needed to reach HCFs, as well as on current inequalities in travel-time. The effect will depend upon the locations and degree to which the healthcare infrastructure is expanded in response to demographic changes in the population. If the expansion is predominantly in urban areas, inequalities in travel-time between rural and urban areas may increase. If the expansion is predominantly in rural areas, inequalities in travel-time between rural and urban areas may decrease. However, HIV treatment programs in Africa are beginning to move away from HCF-based programs to non-HCF based programs[59,60]; exactly what type of programs should be implemented, and their cost-effectiveness, are currently being evaluated. The impact that these non-HCF programs will have on decreasing inequalities in access to HIV treatment will depend upon what percentage of the HIV-infected population is served by these programs, and where these programs are implemented.

There are several limitations with the analyses that we have conducted. First, the PHIA travel-time data were only coarsely defined (there were only

**Fig. 4 | Travel-time inequities. a** National-level travel-time differences between Eswatini (green), Malawi (pink) and Zambia (purple). Urban (red) and rural (blue) differences for **b** Eswatini, **c** Malawi, and **d** Zambia. Dots show the cumulative proportion of individuals traveling for treatment for: one, two, and either 3.5 hours (urban areas) or four hours (rural areas). These are derived from the self-reported travel-time data. Fitted curves are logistic CDFs; shaded regions show 95% confidence bands for these curves.

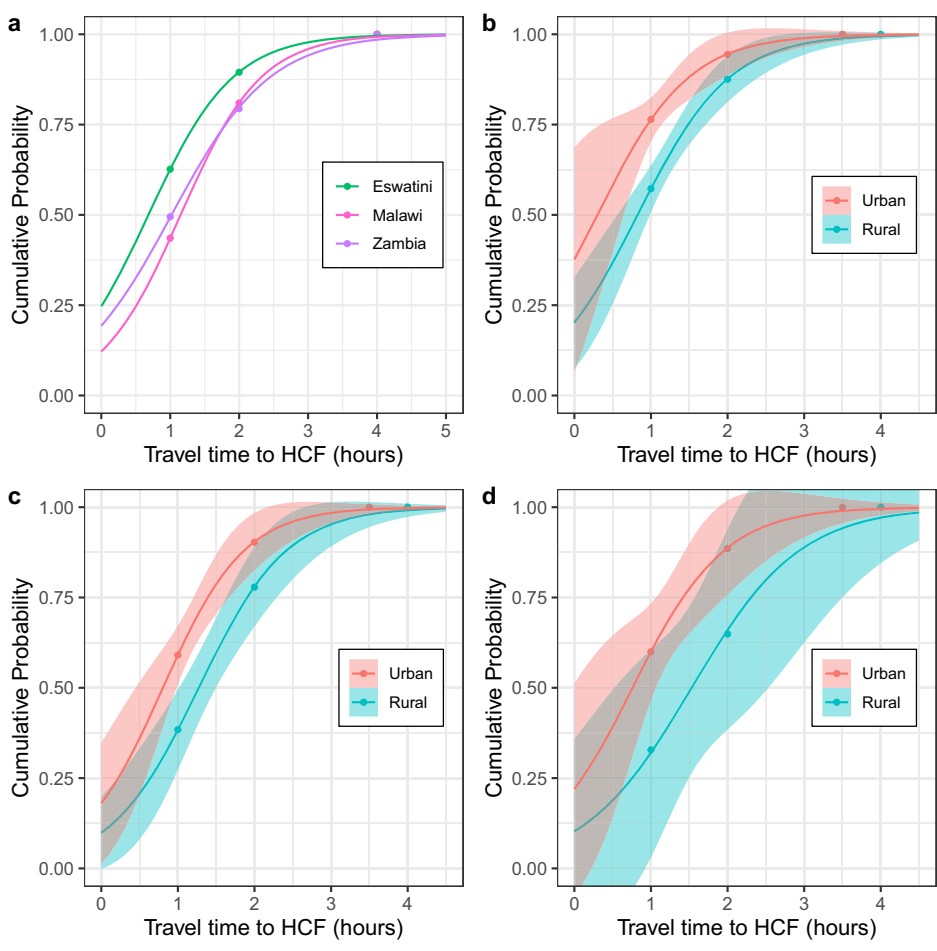

### Table 3 | Inequities in travel-time to access ART amongst and within Eswatini, Malawi, and Zambia

| Travel-time to ART | Eswatini | | | Malawi | | | Zambia | | |
|---|---|---|---|---|---|---|---|---|---|
| | Overall | Rural | Urban | Overall | Rural | Urban | Overall | Rural | Urban |
| One or more hours | 37.4% | 42.8% | 23.6% | 56.4% | 61.6% | 40.9% | 50.5% | 67.2% | 40.0% |
| More than two hours | 10.5% | 12.5% | 5.6% | 19.0% | 22.1% | 9.7% | 20.6% | 35.1% | 11.5% |

The percentage of PWH on ART with a one-way travel-time of (i) one or more hours, and (ii) more than two hours, by country and urban-rural residence.

### Table 4 | Odds of traveling for one or more hours to access treatment

| Effect | Eswatini | | | Malawi | | | Zambia | | |
|---|---|---|---|---|---|---|---|---|---|
| | aOR | 95% CI | *P* | aOR | 95% CI | *P* | aOR | 95% CI | *P* |
| Rural | 2.36 | (1.89, 2.96) | <0.001* | 2.22 | (1.81, 2.72) | <0.001* | 3.32 | (2.69, 4.11) | <0.001* |
| Women | 1.28 | (1.05, 1.57) | 0.014* | 1.17 | (0.93, 1.47) | 0.192 | 1.74 | (1.39, 2.20) | <0.001* |

Results from country-specific multivariable logistic regression models, presented as aORs with 95% CIs. Hypothesis tests were two-tailed and evaluated at α = 0.05; significant effects are denoted with an asterisk (*).

three categories) and right-censored; therefore, we do not know the maximum travel-time for those who traveled more than two hours. Nevertheless, we found substantial differences in the travel-time amongst, and within, countries. Previously, using a different data set for Malawi, we modelled travel-times for all PWH in the country (i.e., for PWH on treatment and for PWH who were not on treatment). We found that the maximum travel-time to the nearest HCF is ~4 hours if driving or biking was possible[47]. These results indicate that, at least in Malawi, inequities in travel-time may be even greater than we have found in this study. Second, given that we used self-reported data on travel-times, we could only determine

CDFs for travel-time for PWH who had been diagnosed and were on treatment: however, at the time of the surveys, many PWH were not on treatment: 23.4% in Eswatini, 30.0% in Malawi, and 37.4% in Zambia. It is possible that some of these individuals may be living in remote areas that are far from HCFs; therefore, when they go on treatment, they may have to spend a long time travelling to reach HCFs. Consequently, increasing treatment coverage may exacerbate current travel-time inequalities. Third, individual questionnaire response rates were about ~10% higher in women. This could be due to men traveling more for work and being less likely to be home when the survey was implemented; men have been found to be less

likely to participate in national HIV surveys than women[61,62]. If these men are more likely to be infected with HIV, or less likely to be aware of their infection or to access treatment, this could lead to biased estimates. Fourth, our findings that HIV prevalence in men is greater than HIV prevalence in women in certain areas of Malawi and Zambia should be interpreted with caution. This is because they are based on calculating the difference between two spatially interpolated surfaces; they are not based on a statistical test. Finally, there are many demographic and socio-behavioral determinants of healthcare utilization that could lead to inequities in accessing ART; here we have only looked at inequities in the geographic accessibility of HCFs. We plan to investigate socio-economic determinants in future studies; in particular, we will focus on elucidating the relationship between poverty and inequalities in travel-time to receive ART.

UNAIDS' new strategy[1] for controlling the HIV pandemic states a vision for reducing inequalities in the push towards its 2030 testing and treatment targets. The targets are to diagnose 95% of PWH, treat 95% of the diagnosed, and achieve viral suppression in 95% of treated individuals by 2030. Our results have significant implications for UNAIDS' HIV health policy. They show that, to achieve UNAIDS' goal, it will be necessary to eliminate inequities in travel-time to ART both amongst – and within – countries. The inequities that we have found were occurring when ART coverage levels were moderate to fairly high. As coverage has increased, it is important to determine whether these inequities have increased (or decreased), to identify the drivers of inequities, and to design strategies to minimize them. Several approaches can be taken to decrease travel-time to ART; these include building new healthcare facilities (as the Government of Malawi plans to do)[63], increasing the availability of transportation[47] (as many PWH have to walk to access ART), and using non-facility based strategies for distributing ART (e.g., client-centered differentiated service delivery models)[59,60]. We have investigated the situation in three SSA countries with HIV epidemics: all have inequities in travel-time to access ART. It is important to determine whether inequities exist in other countries that are afflicted with HIV. Such investigations are critical in order to enable the effective targeting, and equitable allocation, of limited resources.

## Data availability
All PHIA survey data used to conduct this analysis are freely available to registered users at the PHIA project website: https://phia-data.icap.columbia.edu/. All source data are freely available in a Dryad repository: https://doi.org/10.5061/dryad.qjq2bvqrz.

## Code availability
All R (v. 4.1.2)[33] and Stata (v.16)[31] code used in this analysis is freely available in a Zenodo repository: https://doi.org/10.5281/zenodo.14903110[64].

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

## Acknowledgements
We acknowledge ICAP at Columbia University for collecting the PHIA data. J.T.O., L.D., W.M. and S.B. acknowledge the financial support of the National Institute of Allergy and Infectious Diseases, National Institutes of Health grant R01 AI167713. We are grateful to Nelson Freimer for discussions throughout the course of this research.

## Author contributions
J.T.O. and S.B. designed the project. J.T.O. and A.L. conducted all data analysis, and accessed and verified the underlying data. L.D., W.M. and H.N.B. provided in-country expertise. All authors contributed to interpreting the results. S.B. wrote the first draft of the manuscript; all co-authors contributed to writing subsequent drafts. All authors read and approved the final manuscript.

## Competing interests
The authors declare no competing interests.
