## [Transparent Peer Review file · Communications Medicine]

Analysis of travel-time to HIV treatment in sub-Saharan Africa reveals inequities in access to antiretrovirals

Corresponding Author: Dr Sally Blower

Version 0:

Reviewer comments:

Reviewer #1

(Remarks to the Author)

Thank you for this interesting manuscript on travel time to access ART. The manuscript is clear and well presented, and presents interesting results on differences in travel time between and within Eswatini, Malawi and Zambia. Below I provide some suggestions for manuscript, starting with overall comments followed by line specific suggestions.

In the results, the authors place much emphasis on the higher prevalence among women compared to men; this is well established. Although the maps are fine to present by sex to compare geographic spread across the sexes, the emphasis on this sex difference in the first paragraph of the results and the discussion are not new and should be reduced. Rather, the emphasis should be on rural/urban difference stratified by sex.

In the results, the authors present the % of people who travelled for over 2-hours to access ART, but the logistic regression model focuses on those who travelled for one hour or more; as such, lines 228 to 231 of the results should describe the % travelling for one hour or more to be consistent across the results.

In the discussion, line 288-291, I don't think women would be more likely to demonstrate bypass behaviour. It is well established that women are more likely than men to access healthcare services in general, including for HIV testing and ART. It is likely that women may need to travel more they are more likely to reside in rural areas (with men more mobile for work). With two countries having predominantly rural epidemics and women more likely to be living with HIV and on ART, longer travel times are likely due to more women living with HIV and on ART in rural areas.

Travel times are high for those residing in urban areas (with more uncertainty around estimates for rural areas, particularly in Zambia). With growing levels of urbanization in many countries, including the three included in this analysis – the authors could discuss more the implications of their findings on urban areas in particular. Can Figure 4 be stratified by sex also for each country?

Abstract

Inequities in travel time is mentioned in the abstract, but inequities by what? Urban/rural living status is the key factor of interest, so make this clear in the abstract.

Introduction

Line 104, change to rural and urban settings

Line 110, I think the words "both time and financial costs" are redundant as it's implicit that these two factors will have a greater impact based on the start of the sentence

Lines 111-112 are repetitive with earlier sentences on the association between rural residence and coverage of HIV testing

Line 114, lost to follow up – be more specific, lost from care?

Line 116, likely that men have more economic ability to access care, regardless of distance; something to add to this final paragraph

Perhaps this journal has a different style, but in public health we include the aim of the analysis in the final paragraph; i.e. move the questions from lines 86-88 to this final paragraph.

Results

Line 184 – 185, make clear you mean HIV prevalence and state: the HIV prevalence was similar in Malawi Zambia

Line 185, where it states: Notably, in all three countries... I don't think the Notably is necessary, it is well established that HIV prevalence is higher in women than men in these countries.

Line 192 – 193, present the prevalence by rural areas for all three countries; at present its urban for Zambia and rural for Eswatini and Malawi

Line 271 – change to: Substantial inequities in travel-time to access ART were found between and within the three countries.

Line 228, don't need to state that rural areas had more travel time – this is implicit based on the first part of the sentence

Line 228-230: add the % for this sentence.

For the OR presented, are these adjusted for age and sex? Not clear from the methods. Add the p-values where the OR are presented.

Line 234 and elsewhere, change “an hour” to “one hour” to be clear.

Figure 3 Spell out DoI acronym in the title and specify units in the legend

Reviewer #2

(Remarks to the Author)

Using data from nationally representative surveys, the paper aims to identify inequities in travel time to HIV treatment. This is an important contribution with many significant implications. The link with biometric data is particularly novel and significant.

The manuscript should be further improved by considering and discussing the limitations, revising descriptions of methods and statistics, incorporating relevant background, and improving the figures for readability.

Please see some specific questions and comments below.

Limitations:

-Self-reported data. It is mentioned that the surveys are nationally representative. Are they representative in terms of gender and rural-urban distribution? Understanding whether this is true is key to estimating the validity of the findings. Different methods and different assumptions when calculating travel times can significantly alter the estimates and the subsequent findings [3], so at least incorporating more details about the background would be useful.

-Contextualization. Similarly, in Line 220, it would be useful to contrast the produced estimates to already established estimates of travel time to healthcare in general [1,2] for additional reference/context. I.e., is X% accessibility in under an hour surprising or aligned with existing estimates?

Statistics

-Supplementary Table 1: What precisely do the rows 7 and 8 refer to? By definition, 90% of points should fall within the 90% confidence interval. How are these numbers calculated? Why not provide confidence intervals of the mean?

-Line 163: It is necessary to explain in sufficient detail how the method works instead of relying on how it was done in previous work.

-Confounding factors. Results (line 186): Could you please consider and discuss the ways in which alternative hypotheses can explain discrepancies between men and women due to the biases in the studied data? E.g., varying rates of awareness of the status, response biases, etc.

Background/literature

-Line 76: Could you outline what would be expected based on the existing literature on the topic of inequalities in access to healthcare (not necessarily limited specifically to HIV-related care, e.g. [1,2])? That would help to outline the research gap better and specify the expectations based on what is known already.

Figures

-Figures are very compelling and easy to understand. Figure 2: It would be useful to display the difference between men and women directly (in a third row) instead of relying on the reader to compare visually.

The construct of gender

-Could you provide additional details about how gender is collected in the surveys (specific categories)?

Other:

-The abstract (in the conclusion) needs to clearly communicate to the reader the limits to the validity of the contributed findings and potential limitations.

[1] Weiss, D. J., et al. "Global maps of travel time to healthcare facilities." *Nature Medicine* 26.12 (2020): 1835-1838.

[2] Gligorić, Kristina, et al. "Revealed versus potential spatial accessibility of healthcare and changing patterns during the COVID-19 pandemic." *Communications Medicine* 3.1 (2023): 157.

[3] Hierink, Fleur, et al. "Differences between gridded population data impact measures of geographic access to healthcare in sub-Saharan Africa." *Communications medicine* 2.1 (2022): 117.

Reviewer #3

(Remarks to the Author)

In their manuscript, Okano et al. assessed the drivers of travel time taken by PLHIV to access ART at health facilities. The results add to the knowledge base in this area of research, and provide additional context for Eswatini, Malawi and Zambia. Please find below some comments and questions about the work presented.

General:

1) What is the definition of PLHIV in this study? Are these people who tested positive for HIV in the PHIA? If yes, what about people with an undetectable viral load, particularly those who have been on ARVs? It's also unclear how this phenomenon would affect the prevalence/coverage estimates.

2) In the PHIA data, is there indication of what type of healthcare facilities people seek ARTs from? (e.g. public or private facilities) It would be a nice addition to include this breakdown and see how it compares across countries.

3) Was there consideration for use modelling techniques to calculate travel time to the health facilities (for example using MAP friction surfaces) rather than self-reported travel time? If so, what were there barriers to, or decisions against this approach? Might be worth explaining in the text why self-reported travel time was used instead of calculating this variable.

Figures/Tables:

4) Figure 4: Is it possible to improve the transparency of the 95% CI bands so that the red band is not obstructing the blue band.

5) It would be useful to have many of the results in the results section in a table, or tables (e.g. the proportions and Odds Ratios, significance tests) for easier side-by-side comparison of the results.

Line-by-line comments/questions:

6) Line 50-51: Could you state the actual travel times?

7) Line 82: PHIA surveys have been conducted

8) Line 99: Please contextualize what 'lost to follow-up' refers to in this sentence – testing or adherence to ART or both, or something else?

9) Line 100-104: Please clarify what 'distance' refers to in these sentences. Is this Euclidean distance, or travelled distance, or travel time to a health facility, or something else?

10) Line 114: Please contextualize what 'lost to follow-up' refers to in this sentence – testing or adherence to ART or both, or something else?

11) Line 126: Please say more about the sampling design. What's stage one, and what's stage two?

12) Lines 139 – 142: Please say whether the entire sample size was used, or (if applicable) whether there were there any observations excluded (for example due to missing data?)

13) Lines 184 - 185: please include confidence intervals or some form of significance metric for these estimates

14) Line 205 – 206: please specify whether these differences in coverage were significant.

15) Line 248: could be better phrased as: 'However, Eswatini and Malawi have a predominantly rural epidemic, while 249 Zambia has a predominantly urban epidemic'

16) Line 253: Please expound on why a low DoI would result in longer travel time to access ART.

17) Line 254: It is misleading to state that 'this is the first study to show that there are inequalities among and within countries in the travel-time needed to access ART'. This is a very broad statement which ignores even some of the studies referenced in this paper.

18) Line 255: Please expound on why this result is unsurprising.

19) Line 306: Please give examples of what other determinants are being planned for further investigation

Reviewer #4

(Remarks to the Author)

The article by Okano et al. examined inequalities in travel time to access ART for people living with HIV in three sub-Saharan African countries. The article is generally clearly written and the effort is commendable. The analytic procedure is equally well stated. However, I have the following concerns regarding the framing of the background and results:

I will think the introduction will need some tweaking. The first paragraph introduces the purpose of the study followed by some kind of literature review but I would think that the review should lead to the problem followed by the purpose of the study. It flows better this way.

I have issues with the presentations of some of the results. The authors assumed the readers can understand some things that are not obviously presented. For instance Figure 1 presents maps for the three countries showing the epidemic prevalence without including the administrative divisions of the countries yet, the authors presented the results as having "large-scale geographical patterns in prevalence that cross administrative boundaries, and there is considerable geographic variation: 14–53% in Eswatini, 1–26% in Malawi, and 1–32% in Zambia ..." The crossing of administrative boundaries is not in any way obvious. I wonder why they three maps are not plotted on the same scale rather than just the later two. Also, the statement that "Gender-stratified ESP maps (Fig. 2) show that the geospatial patterns in prevalence differ between women and men, and prevalence is generally higher in women than in men" does not appear to hold true everywhere across the

spatial domains as there are places where they are comparable, for instance in Malawi. For Figure 3, I couldn't differentiate between rural and urban as presented in lines 201-203 except if I am missing out something. I am assuming the results presented in the third paragraph beginning from line 205 are descriptive findings. If so, they should better come before those from the model.

Line 288 in the result section, I couldn't get what the statement "The differences between the genders is not obvious" in the light of the preceding lines.

Reviewer #5

(Remarks to the Author)

Brief summary of research

Okano et al present an analysis of travel time (one way) for people living with HIV to access antiretroviral therapy in three countries: Eswatini, Malawi, and Zambia, using the Population-based HIV Impact Assessment (PHIA) surveys. Country-specific data were used to estimate HIV prevalence and treatment coverage. Prevalence maps were constructed using spatial interpolation (Bayesian kriging), which were then used to create Density of Infection maps which estimate the number of PLHIV per each square kilometer. Travel-time estimates were based on reported travel-time to healthcare facilities to access treatment: <1 hour, 1-2 hours, or >2 hours which were used to fit logistic cumulative distribution functions and logistic regression models. Inequities in travel time as well as differences in PLWH demographics and treatment coverage by country were presented in the results.

Overall impression

The authors should be commended for this rigorous assessment of an important metric (i.e., travel time to a healthcare facility) that factors into realizing the UNAIDS goal of ending inequalities in accessing HIV treatment. Real-world data on travel time is limited, so use of population-based surveys to develop some insights into this problem can certainly help inform policies and interventions aimed at increasing ART coverage. The combination of surface prevalence maps and DoI maps are an innovative and useful way to estimate the geographic locations of people living with HIV who would need to be on treatment to achieve the country's current treatment coverage. Travel time estimates stratified by urban/rural and women/men highlight inequities between different segments of the population. This is a useful contribution to the literature, utilizes several innovative geospatial techniques applied to population based surveys, and has the potential to inform real-world policy making.

Specific comments: Major and minor issues

1. Methods, line 147-148. Authors state that they did not create treatment coverage maps...as the treatment data were too sparse to construct such maps." As a reader, this made me wonder exactly how this is a limitation and if there is a certain threshold needed to create 'surface prevalence maps'.
 - a. Recommendation: Please clarify in more detail why treatment coverage maps were not created in the same manner as the prevalence maps.
2. Results, line 195-196. Authors report that geographic patterns in prevalence cross administrative borders. The accompanying figures do not show any subnational (admin level 1 or 2) boundaries.
 - a. Recommendation: Maps could be improved by displaying the subnational boundaries if the claim is geographic patterns cross administrative borders.
3. Overall, the results were presented in a clear and easy-to-understand order, with useful data points highlighted appropriately.
4. In Figures 1 and 2, it's not clear to me if the second color scale (values ranging 0 - 0.34) should be used to interpret the maps of both Malawi (b and e) and Zambia (c and f), or does it only refer to Malawi (b and e), and there's a missing color scale for Zambia (c and f).
 - a. Recommend: Creating a third color scale so it is clear that each country has its own color scale to use in interpreting the map.
5. Figures 1, 2, and 3. These maps can also be improved by displaying the capital or major cities for reference.
6. Discussion, line 306-307. With regards to self-reported travel-time data, there has been a recent study analyzing reported versus modelled travel-time, and the discrepancies between the two.
 - a. Recommendation: please refer to Bouanchaud et al. BMJ Global Health 2022 (<https://doi.org/10.1136/bmjgh-2021-008366>).
7. One suggestion, not necessarily for this manuscript, would be to construct DoI maps that reflect where geographically you would need to target for increasing treatment coverage to meet certain targets, for example 95% of people on treatment in a given country.

Version 1:

Reviewer comments:

Reviewer #1

(Remarks to the Author)

Thank you to the authors for the revised manuscript; the manuscript has improved in line with the changes. Below are a few final suggested changes. These are predominantly minor, except perhaps the suggestion to include a Table 1 describing the

populations included in the study by country.

The results still place a lot of emphasis on sex differences. For example, line 220 states that HIV prevalence was generally higher among women than men. This isn't a new finding, and again should be premised with "as expected" or similar. The results report that in some areas HIV prevalence is higher in men than in women. While this could be the case, no confidence intervals are reported and response rates were higher among women than men, so these findings should be interpreted with caution. This is mentioned in the discussion, but it could be made more explicitly related to these findings.

Where the authors state that there was a "significant difference" (e.g. line 230), the p-value should be included.

The authors should include a Table 1 that describes the populations included in the study; how many males/females, age distribution, how many resided in urban/rural areas. This would help the reader contextualise the findings.

In line with Reviewer 5 comment, the legend for maps 2 is still unclear; it might just be a case of moving the legend for Malawi and Zambia such that it is to the right of Zambia.

In line 368-369 of the discussion, it states that PHW were asked if they travelled more than 2 hours to access ART. This raises the question, why did the authors present results for one hour? Perhaps rephrase this part of the discussion such that the question is described as presented in the methods.

Reviewer #2

(Remarks to the Author)

I thank the authors for their detailed revisions. My concerns have been addressed. I have no further comments.

Reviewer #3

(Remarks to the Author)

Thank you to the authors for providing the revised manuscript for further comments. Please find below some additional suggestions for your consideration.

General comment -> please consider reporting prevalence/proportions with greater precision (e.g. at least 1 decimal point) throughout the text.

Lines 51 and 66 -> 'Some patients travelled (one-way) for less than one hour...' is vague. Please include proportions or rephrase this sentence. Perhaps focus on what the most prevalent travel times were (overall or by country)

Line 131 -> It's not clear to me what the phrase 'HIV status was identified by using a serological rapid diagnostic testing algorithm that was based on each countries' national guidelines' means exactly. Are these guidelines different between countries? Are different RDTs used for this diagnosis across the countries? If there is a difference in this diagnosis or guidelines, does this impact interpretation of results between countries? Either way, because the authors have alluded to a difference, it would be worth clarifying if this difference is important in interpretation of results - perhaps in the discussion section.

Line 133-> should be "each country's" instead of "each countries".

Table 1 -> please include the sample sizes for prevalence and treatment coverage proportions

lines 232-245 -> some of these descriptive proportions for the sampled populations (eg male/female; urban/rural; less than one hour/ one to two hours / more than two hours) would be better represented in a table or figures for easier comparison across countries.

Lines 347 -> this could be presented more clearly in the results section if the authors choose to take my suggestion on lines 232-245 stated above.

Lines 376-377 -> please consider putting this in the results section.

Figure 4-> please include a legend which explains the colour coding for these graphs in the figure. Secondly, there is quite an overlap of Cis in figures 4b-d, particularly for Zambia. Could you say more about this in the discussion?

Line 382 - for the statement 'men 383 are also more likely to refuse to participate in surveys than women.' Is there a reference for this? If not, please consider rephrasing to indicate that this is a hypothesis.

Conclusion-> please consider having a clearly marked conclusion section.

Reviewer #5

(Remarks to the Author)

Thank you to the authors for responding in a thoughtful manner to the reviewer comments. I felt that the edits made were reasonable and contributed to a clearer manuscript, specifically including the revisions to the figures and figure legends. This appears to be a significantly improved manuscript.
